# Malignant and Benign T Cells Constituting Cutaneous T-Cell Lymphoma

**DOI:** 10.3390/ijms222312933

**Published:** 2021-11-29

**Authors:** Shuichi Nakai, Eiji Kiyohara, Rei Watanabe

**Affiliations:** 1Department of Dermatology, Course of Integrated Medicine, Graduate School of Medicine/Faculty of Medicine, Osaka University, Osaka 565-0871, Japan; 149nakai@derma.med.osaka-u.ac.jp (S.N.); eiji-kiyohara@derma.med.osaka-u.ac.jp (E.K.); 2Research Department, Maruho Co., Ltd., Kyoto 600-8815, Japan; 3Department of Integrative Medicine for Allergic and Immunological Diseases, Course of Integrated Medicine, Graduate School of Medicine/Faculty of Medicine, Osaka University, Osaka 565-0871, Japan

**Keywords:** cutaneous T-cell lymphoma, mycosis fungoides, skin resident memory T cells, malignant T cells, benign T cells

## Abstract

Cutaneous T-cell lymphoma (CTCL) is a heterogeneous group of non-Hodgkin lymphoma, including various clinical manifestations, such as mycosis fungoides (MF) and Sézary syndrome (SS). CTCL mostly develops from CD4 T cells with the skin-tropic memory phenotype. Malignant T cells in MF lesions show the phenotype of skin resident memory T cells (T_RM_), which reside in the peripheral tissues for long periods and do not recirculate. On the other hand, malignant T cells in SS represent the phenotype of central memory T cells (T_CM_), which are characterized by recirculation to and from the blood and lymphoid tissues. The kinetics and the functional characteristics of malignant cells in CTCL are still unclear due, in part, to the fact that both the malignant cells and the T cells exerting anti-tumor activity possess the same characteristics as T cells. Capturing the features of both the malignant and the benign T cells is necessary for understanding the pathogenesis of CTCL and would lead to new therapeutic strategies specifically targeting the skin malignant T cells or benign T cells.

## 1. Introduction

Cutaneous T-cell lymphoma (CTCL) is a heterogeneous group of non-Hodgkin lymphoma. The clinical manifestation of CTCL is diverse. The most common type of CTCL is mycosis fungoides (MF), presenting with inflammatory skin lesions, such as erythematous patches, plaques, and tumors infiltrated by both malignant and benign T cells [1,2]. The prognosis of early-stage MF is good, and skin-directed therapies can manage the disease activity for the long term in many cases [3]. On the other hand, the advanced-stage MF with the development of tumors, erythroderma, and involvement in the lymph nodes is regarded as showing a poor prognosis. It is thus important to control the disease activity of the early-stage MF to prevent the progression to the advanced stage. Another disease subtype, Sézary syndrome (SS), which is a rare and more aggressive type of CTCL, presents erythroderma, lymphadenopathy, and blood-circulating malignant T cells called Sézary cells from the early phase of disease [1,4]. Malignant T cells in CTCL mostly develop from CD4 fraction and possess the skin-tropic memory phenotype, and the lesions are regarded as primarily developing in the skin [1]. The elucidation of molecular and cellular biology in CTCL remains incomplete due, in part, to the fact that malignant T cells and non-malignant infiltrating T cells are both confined in the same lesional sites. However, recent advances in the next-generation sequencing approaches are adding drastic suggestions concerning the disease pathogenesis.

Skin is a large barrier tissue which serves to prevent foreign antigens from entering the body. Skin serves as both a structural and an immunological barrier, and healthy human skin contains an estimate of 20 billion memory T cells [5]. Over half of these skin T cells are understood to remain in the skin for a long period without recirculating to and from blood and lymphoid tissues, and this subpopulation is now called resident memory T cells (T_RM_) [6,7]. T_RM_ provide a stronger local adaptive defense compared to circulating memory T cells [8,9,10,11], and they can exert a sufficient response to the local antigen re-exposure without the aid of circulating T cells [12]. In addition to the function as a local defense against antigens, recent studies suggest that T_RM_ also provide a systemic response upon re-exposure to antigens by proliferating and baring circulating memory T cell populations [13,14]. Besides infectious diseases, the involvement of skin T_RM_ is now recognized in many cutaneous conditions, such as allergic contact hypersensitivity [15], chronic immune-mediated inflammatory diseases, including vitiligo and psoriasis [16,17,18], fixed drug eruption [19], and cutaneous malignancies [20]. The engagement of malignant and benign T_RM_ in the pathogenesis of CTCL is also being elucidated [7,21,22].

In this review, we first describe the property of skin T_RM_, and then provide the characteristics of malignant and benign T cells in MF/SS from the aspects of T_RM_ and T-cell phenotypes. We also mention the oncogenic mechanisms and the tumor microenvironment which would affect both malignant and benign T cells in CTCL.

## 2. The Development of Skin T_RM_

Only a few T cells exist in newborn human skin [22], and the population of T_RM_ is presumed to be built by the recruitment of circulating T cells according to the repeated exposure to various antigens. The global characteristic of T_RM_ is tissue retention, and this property can be developed by complex factors, including cytokines, chemokines, their receptors, other cell-surface molecules for tissue homing and retention, and transcription factors.

The cell-surface molecules CD69 and CD103 are the most frequently used markers for recognizing T_RM_. CD69 interferes with sphingosine-1-phosphate (S1P) receptor-1, which senses the density gradience of S1P and helps the cells to exit from peripheral tissues to lymphoid organs and blood [23,24]. CD103 is a ligand of E-cadherin which is expressed on epithelial cells [22,25]. However, the existence of T_RM_ without the expression of CD69 and/or CD103 has been reported [26,27], and T_RM_ with CD103 expression can also be found in the sites which lack E-cadherin expression, such as the dermis and central nervous system [22,28]. Thus, these two molecules are not the universal markers for T_RM_.

T_RM_ can also be identified by the expression of transcription factors. For instance, skin T_RM_ highly express the aryl hydrocarbon receptor (AhR) compared with naïve T cells and splenic T cells, and AhR presumably contributes to the long-term persistence of epidermal T_RM_ [29]. As for the T_RM_ in other tissues, the maintenance of intestinal CD4 T_RM_ may be related to Hobit and Blimp-1, and the deletion of these molecules results in functional impairment of CD4 T_RM_ in the murine model of inflammatory bowel disease (IBD) [30]. Furthermore, circulating effector T cells expressing Hobit are identified as T_RM_ precursors, which preferably form CD8 T_RM_ during antigen exposure [31]. These findings might be adapted to skin T_RM_ too. The upregulation of Notch [32], Hypoxia-inducible factor-1α [33], Runx3 [34], and basic helix-loop-helix family member E40 [35] have also been reported to be involved in the differentiation and/or maintenance of T_RM_. However, as, for instance, lung CD8 T_RM_ are shown not to rely on Hobit expression [36], it is also possible that the transcription factors involved in the development and persistence of T_RM_ vary among tissues.

While skin T_RM_ has a common property with the T_RM_ in other organs, they are shown to express distinct molecules related to skin tropism. For instance, CCR6 is highly expressed by CD8 T_RM_ in psoriasis lesions [16,17], and CCR6^+^ CD8 T_RM_ precursors possibly enter the skin according to the concentration gradient of the ligand CCL20 that is upregulated in psoriatic keratinocytes [37]. The chemokine receptors, such as cutaneous lymphocyte-associated antigen [5], CCR4 [38], CCR8 [39], CCR10 [40], CXCR3 [26,37], and CXCR6 [41] have also been reported to play important roles in homing and/or retention in skin. The contribution of some of these molecules to MF pathogenesis and prognosis has been demonstrated [7,42,43], supporting the skin-tropic phenotype of malignant T cells in MF and the involvement of the tumor microenvironment in disease manifestation, as described in Section 7. However, although the malignant T cells in CTCL are in most cases CD4 T cells, the information on the development of CD4 T_RM_ is still limited.

IL-7 and IL-15 are major cytokines that enable T_RM_ to stay in the skin for the long term [44]. IL-7 binds to a heterodimer of the IL-7 receptor α (IL-7Rα, also named CD127) and IL-2Rγ (also named CD132) and is involved in T-cell survival and proliferation via downstream molecules, such as Janus kinase (JAK) 1, JAK3, and phosphoinositide 3-kinase (PI3K) [45]. IL-7 is also required for the development [46] and maintenance [47,48] of memory T cells. The receptor of IL-15 is a heterodimer of IL-2Rβ (also named CD122) and CD132 [49]. IL-15 signaling is also transmitted via JAK1 and JAK3 and leads to the promotion of T-cell survival, proliferation, and cytokine production. IL-15 supports the generation of memory T cells from naïve T cells and enables memory T cells to proliferate rapidly in response to antigen re-exposure [50,51]. In the skin, the major sources of IL-7 and IL-15 are fibroblasts and keratinocytes, and the upregulated production of IL-7 is reported in the hair follicles of both MF and SS [52,53], suggesting the role of IL-7 in the recruitment, survival, and proliferation of malignant T cells in CTCL lesions. Deletion of IL-7 or IL-15 attenuates the skin inflammation caused by CD8 T_RM_ in the murine contact hypersensitivity model [44], and administration of the CD132-neutralizing antibody decreases the number of CD4 T_RM_ in the murine allergic airway models [54]. Besides, the CD122-neutralizing antibody reverses the diseases of a murine model of vitiligo, a chronic skin depigmenting disorder, by the suppression of CD8 T_RM_ [55]. As JAK1, JAK3, and the signal transducer and activator of transcription (STAT) are the overlapped molecules which are involved in the signaling pathways of IL-7 and IL-15, they have been hypothesized to play important roles for T_RM_ maintenance [56]. At the same time, considering the recent study demonstrating that the development of skin CD8 T_RM_ is not efficiently impaired by the administration of a JAK inhibitor in a murine vitiligo model [57], JAK-independent signaling pathways of IL-7/IL-15 are also suggested to be involved in the construction of T_RM_. As for CTCL, the activation of the JAK/STAT pathways leads to MF progression accompanied by the upregulation of the cell cycle in malignant T cells [58,59]. The suppressive effects of JAK inhibitors are also reported in the proliferation of CTCL cell lines and SS PBMC T cells [60,61].

## 3. The Function of Skin T_RM_

Among T_RM_, the CD8 fraction has been clarified more thoroughly compared to the CD4 fraction. Just like the general effector CD8 T cells, CD8 T_RM_ can be classified into Tc1, Tc2, Tc9, Tc17, and Tc22 by the cytokines they produce [62]. Besides their roles in the defense against pathogens in the barrier tissue, their functional characteristics are known to be related to some inflammatory skin disorders and tumor immunity. For example, in the pathogenesis of vitiligo, IFNγ-producing CD8 T_RM_ has been shown to play a major role [16,55,63]. CD49a, which binds to collagen IV in the basement membrane, is reported to be expressed by CD8 T_RM_ in vitiligo lesions [16]. CD49a^+^ CD103^+^ CD8 T_RM_ are localized to the epidermis, are excelled at IFNγ production, and rapidly gain a cytotoxic property in response to IL-15 stimulation. These T_RM_ are also involved in alopecia areata (AA), and the granzyme B production from them is related to treatment resistance [64]. Thus, the CD8 T_RM_ found in vitiligo and AA are characterized as Tc1-like cells, whereas in the lesional skin of psoriasis, CD8 T_RM_ express IL-17A, IL-22, and IFNγ [17,65,66]. IL-17A-producing CD8 T_RM_ show a high expression level of CCR6, IL-23 receptor, and/or CD49a [16] and remain in the cured sites for the long term [17].

The investigation on the contribution of CD8 T_RM_ to the tumor immunity is also progressing rapidly from the aspect of tumor-infiltrating lymphocytes, and tumor-engrafted murine models have revealed the antitumor function of T_RM_ in multiple tumor strains [67,68]. In the actual human solid cancer settings, the property of T_RM_ is associated with prognosis. For instance, the infiltration of CD103^+^ CD8 T_RM_ positively correlates with good prognosis in various solid tumors, including breast cancer, esophageal cancer, gastric cancer, lung cancer, and malignant melanoma [69,70,71,72,73,74]. Various subpopulations of the tumor-infiltrating T cells express the T_RM_ marker CD103, possibly reacting with E-cadherin expressed by cancer cells and residing in the tumor [68,75], and the expression of CD103 correlates with the cytotoxic function of these T cells with the production of IFNγ and granzymes [67,68,76]. T_RM_ also facilitate the antitumor immunity by the activation of dendritic cells, natural killer cells, and non-specific T cells via the production of the effector cytokines [32,77,78,79]. The involvement of this population in CTCL is mentioned in Section 6.

Although the characterization of T_RM_ has been clarified more in CD8 than in the CD4 fraction, both in the murine models and in humans, research on CD4 T_RM_ has also progressed in recent years. Compared to CD8 T cells, CD4 T cells in skin generally express less molecules related to tissue residency and are presumed to be more mobile [22,80,81]. At the same time, recent research has demonstrated that CD4 T_RM_ are also involved in immune reactions against pathogens such as mycobacterium tuberculosis, herpes simplex virus-2, and the varicella zoster virus and immune-related disorders, including IBD and allergic asthma [30,82,83,84,85]. The functional characteristics of CD4 T_RM_ varies depending on the disease condition, mostly in concordance with the characteristics of co-existing CD8 T_RM_. The malignant T cells in MF typically arise from the CD4 fraction and possess the T_RM_ phenotype, especially in the early stage [2,7], and the varied properties of CD4 T_RM_ might be reflected in the disease manifestation of MF.

## 4. Distinguishment of Malignant and Benign T Cells in CTCL

In many cases of the early-stage MF, the majority of T cells in the skin lesions are benign T cells [86]. These benign T cells are supposed to exert antitumor immunity against malignant cells as tumor-infiltrating T cells [87]. However, it is difficult to distinguish the benign T cells from the malignant T cells because both share the same nature as T cells. Both malignant and benign T cells include T_RM_ fractions and the definition by T_RM_ markers is impossible [7,22]. There are some surrogate cell-surface markers to distinguish between malignant and benign T cells, such as CD5, CD7, and CD26 [88,89].

Among them, the most frequently used marker in the clinical settings is CD7. Loss of CD7 is regarded as the characteristic property of malignant T cells in the skin lesions of MF [90] and the peripheral blood of SS [91]. CD5 is a scavenger receptor cysteine-rich family transmembrane glycoprotein expressed on all T cells [92]. Antigen-specific T cells overexpressing CD5 reportedly persist better as memory T cells after peripheral activation [93]. The loss of CD5 is sometimes seen in the advanced stages of MF [94]. However, the expression of CD5 is also reported to be higher in the malignant T cells of SS compared to the benign T cells [89]. Although the expression of this molecule is used in the diagnosis of CTCL, combined with the other information, it would be difficult to reach the diagnosis solely depending on CD5 expression. CD26 is a multifunctional type II cell surface glycoprotein widely expressed on a variety of CD4 T cells [95]. Malignant T cells frequently lose CD26 expression in both the skin and the peripheral blood of SS and MF, and thus can be a good surrogate marker for distinguishing malignant and benign T cells [88,95].

Malignant and benign T cells in skin lesions could also be distinguished by the size and complexity of the cells, not only in histology, but also in flow cytometry analysis [2,96]. The large cells with complex nuclear shape are usually regarded as malignant T cells. In practice, however, skin malignant T cells are as small as their benign counterparts in some MF and other CTCL subtypes. The change of the sizes of malignant T cells could also be experienced. The combination of the cell-surface molecules and cell sizes would be helpful for the detailed distinction.

As the malignant cells in CTCL are rather heterogeneous, with multiple mutant subclones in the same lesions [97,98], and exist along with their benign counterparts, it was difficult to identify the common and specific characteristics of the malignant cells. Recently, clarifying malignant T cells and their properties by high-throughput sequencing analyses, including single-cell RNA sequencing, has enabled the discovery of the common gene expression signatures of malignant T cells. For instance, highly proliferating malignant T cells, with the expression of the thymocyte selection-associated, high-mobility group box, share the increased gene expression signature involving cell-cycle progression, proliferation, metabolic processes, and resistance to apoptosis [99,100]. The expression of multiple inhibitory receptors is also confirmed in benign T cells from the advanced-stage MF [100]. While it would be currently difficult to separate the live malignant and benign T cells depending on these intracellular molecules, the further progression in analytical techniques would enable the elucidation of the biological characteristics of both the malignant and the benign T cells in the near future.

## 5. Malignant T Cells in CTCL

The malignant T cells in the early-stage MF skin lesions typically represent the T_RM_ phenotype [7]. In a rare case of CD8 MF, the malignant T cells were also demonstrated to have CD8 the T_RM_ phenotype, characterized by the high expression of CD69 and CD103 and the low expression of CD62L and CCR7 [101]. On the other hand, malignant T cells from the skin and blood of SS patients typically show the T_CM_ phenotype [7,102]. The recirculation pattern of T_CM_ is similar to that of the naïve T cells, and they migrate between the blood and the secondary lymphoid organs [103]. T_CM_ are also understood to enter the peripheral tissues according to their tissue-tropic molecules [104]. The difference in the clinical appearance and the sensitivity to the systemic therapies can partially be explained by this phenotypic difference of malignant T cells in CTCL [22] (Figure 1).

As for the immunophenotype, while the malignant T cells generally represent the Th1 profile in the early stage of MF, they shift to Th2 property according to the disease progression. For instance, in the patch to plaque stage of MF, the ratio of the cells positive for the Th1 master regulator t-box family of transcription factors (t-bet) is higher than that of the positive for the Th2 master regulator gata-binding factor 3 (gata-3). In contrast, gata-3 outdoes t-bet expression in the tumor stage of MF [86]. As the Th2 cytokine, the lesional IL-4 expression, which is comparable with healthy control skin in the early stage, gets higher in the advanced stage of MF [105]. The malignant T cells in SS are also understood to show a Th2 profile with a high IL-4/IL-13 expression and a low TNFα/IFNγ expression [106]. Recent research demonstrates that the phenotypical change of the malignant T cells is affected by the tumor microenvironment (TME), as mentioned in Section 7.

IL-7 is high in the lesional skin of CTCL, both in the mRNA and the protein level, and this skin-derived IL-7 contributes to the proliferation of malignant T cells with high IL-7Rα expression. Another study revealed that IL-15 prolongs the survival of malignant T cells from SS [107]. IL-15 is highly expressed in CTCL lesions and blood, including the T cells [52], and the overexpression of IL-15 in CTCL T cells is reported to be due to the disruption of epigenetic modification [108]. Interestingly, the deletion of IL-7 suppresses the development of the CD4 T_RM_ population in a murine CTCL model, which represents the pathological manifestation mimicking MF, where the epidermotropic T cells show increased IL-7Rα expression [44]. IL-15 transgenic mice also develop skin lesions representing the pathological characteristics of MF [108]. These results imply the importance of IL-7 and IL-15 in the development of malignant T_RM_ in CTCL, especially in MF.

Recently, two models of the T_RM_ cell lineage divergence have been reported [109]. The first model is called the ‘local divergence’ model, which is based on the concept that the circulating memory precursor pool is composed of the cells that are equal in their potential to contribute to both the T_RM_ cell pool and the circulating memory T-cell pool [110,111]. According to this concept, T_RM_ are replenished rather randomly from the memory precursors in circulation by the assistance of the microenvironment, including the enrichment of TGFβ [112,113], IL-7, and IL-15 [44,111,114] signaling. The other model is called the ‘systemic divergence’ model, which is based on the concept that the memory precursor cells are destined to differentiate into T_RM_ or circulating memory T cells within the lymphoid tissues or blood [34,115]. In this concept, the cellular fate of the T_RM_ precursors is already decided at the time of differentiation into memory precursors, possibly shifting to the gene expression profiles similar to those of T_RM_ [115,116] and waiting for the differentiation into T_RM_ until the local inflammation or antigen exposure occurs. The disease course of CTCL, including lesion expansion both in size and in number and the changes of clinical manifestation, and the fact that multiple subclones on the same evolutionary tree of T-cell clones are found in different skin lesions, regardless of the well-demarcated patch/plaques or ill-demarcated tumors, may possibly be explained by the ‘systemic divergence’ model. In other words, parts of the T_RM_ precursors might acquire or already have acquired the malignant property in the lymph nodes or blood before infiltrating into the skin and forming MF lesions. Further studies are awaited as the reliable methods for distinguishing T_RM_ precursors from the other memory T cell precursors have not been established. Revealing the relationship between T_RM_ precursors and CTCL development might lead to the establishment of predicting indexes for disease progression.

## 6. Benign T Cells in CTCL

The non-clonal benign T cells in CTCL lesions are understood to exert antitumor immunity against their malignant counterparts. The early-stage MF lesions contain benign Th1 cells and Tc1 cells [86,117], which are supposed to produce cytotoxic molecules, such as granzymes, perforin, and IFNγ. Thus, benign T cells surrounding CTCL lesions might play a role in suppressing the progression of CTCL. While the expression of T_RM_ markers, especially CD103, in tumor-infiltrating T cells is associated with a stronger antitumor effect in various solid malignant tumors, as described above [69,70,71,72,73,74], benign T cells in CTCL lesions possess less T_RM_ (CD69^+^ CD103^+^) phenotype [97,118]. From the perspective of suppressive functions in antitumor immunity, these benign Th1 cells more frequently express immune checkpoint molecules, such as the programmed cell death 1, lymphocyte activation gene 3 (LAG-3), and cytotoxic T-lymphocyte-associated protein 4 [97]. The benign Tc1 cells also highly express LAG-3, consistent with their less inflammatory phenotype, with the decreased production of cytotoxic molecules [97]. Furthermore, as the disease progresses, CTCL lesions are shifted to the Th2 environment [119]. In skin, Th2 cells are dominated in the dermal fraction with less expression of T_RM_ markers [22]. Th2 cytokines have been reported to show tumor-promoting effects in some solid tumors [120,121,122], and among these cytokines, IL-13 and thymic stromal lymphopoietin (TSLP) are shown to be directly involved in the progression of CTCL. These studies imply that benign T_RM_ in CTCL lesions are decreased both in number and antitumor function with stronger immune-suppressive property. The profiles of benign T cells also change according to the disease progression.

CTCL is treated with various modalities, including topical therapies, ultraviolet (UV) therapies, radiation therapies, and systemic therapies [94]. It has been reported that the cytokine profile of benign T cells is significantly changed before and after effective treatments. While IL-4 production is dominant in benign T cells before treatments, it starts producing IL-2, IFNγ, and TNFα after treatments such as UV, alemtuzumab, and gemcitabine [106]. It has also been revealed that the expression of Th1-related genes, including CXCL9, CXCL10, and CXCL11, correlates with the number of benign T cells in the cured lesions after psoralen and UVA therapy [21], which suggests that the modified expression profile of cytokines and chemokines in the skin microenvironment may lead to the profile shift of benign T cells from Th2 to Th1 after treatments.

A recent study revealed that c-Kit^+^ dendritic cells produce CCL18, which recruits benign Th2 cells, including both T_RM_ and non-T_RM_, and creates an inflammatory synapse among dendritic cells, benign T cells, and malignant T cells [21]. Interactions between the benign OX40^+^ Th2 cells and the OX40L^+^ c-Kit^+^ dendritic cells, and between CD40 on benign Th2 cells and CD40L on malignant T cells, may drive the antigen-independent activation of benign Th2 cells, leading to the visible skin inflammation. This inflammation is diminished after UV therapy, with the loss of correlation between the CCL18 and benign T cells in the CTCL lesions. Considering other studies suggesting that CCL18 expression from macrophages and dendritic cells in the lesions is related to the severity and progression of CTCL [123,124], the benign Th2 cells within CTCL lesions might serve as pro-tumorigenic T cells and might be associated with the tumor progression of CTCL.

## 7. Tumor Microenvironment in CTCL

The difficulty in the clarification of malignant and benign T cells can be caused by the effects of TME on both the malignant and the benign T cells [125,126]. Among the various cells constituting TME, for example, tumor-associated macrophages (TAM) are increased in MF and SS lesions with the increased expression of CCL18 and CCL22, which promote a Th2-biased microenvironment, as mentioned above [123,127]. Indeed, malignant T cells of CTCL highly express CCR4, which is a receptor of CCL18 and CCL22 and also serves as a T_RM_ marker [7,97]. Additionally, in a murine CTCL model, depletion of TAM results in a decrease in tumor size and an increase in tumor-infiltrating CD8 T cells and in the level of antitumor cytokines [128].

Cancer-associated fibroblasts (CAF) isolated from the lesional skin of CTCL are also reported to promote tumor progression and the Th2 environment by the production of various mediators [129,130]. Among these mediators, eotaxins are considered to contribute to tumor migration via interaction with CCR3^+^ lymphocytes and the promotion of the Th2 environment in CTCL [129]. It has also been reported that CAF decrease Th1 chemokine, leading to a Th2-dominant microenvironment, via the production of the herpes virus entry mediator (HVEM) [130]. Another study suggests that CAF protect the malignant T cells from chemotherapy-induced cell death and increase their migration by the interaction of CXCL12 with CXCR4 expressed on the malignant T cells [131]. Actually, normal fibroblasts and CAF show the opposite effects on the CTCL cell line [132]. Normal fibroblasts suppress the expression of MF biomarkers such as *twist-related protein 1* (*TWIST1*) and *thymocyte selection associated high mobility group box* (*TOX*) and enhance the expression of the Th1-related gene, including *IFNγ* and *T-bet*, although CAF promote the expression of these molecules and suppress the expression of the Th1-related gene. These studies suggest that normal fibroblasts can suppress the expression of disease-promoting genes from the malignant T cells in MF and create an unfavorable environment for their proliferation. The indolent disease progression in the early stage of MF may be at least partially explained by these tumor-suppressing effects of normal fibroblasts.

From the aspects of the molecules, the cytokines and chemokines which contribute to the recruitment and differentiation of TME-related cells, including TAM, CAF, and T_RM_ (both benign and malignant), would also be counted as constituting TME. IL-7, IL-15, and TGFβ can develop the population of both the malignant and the benign T_RM_ [52,107,108,112,113], and TGFβ helps the induction of TAM and CAF [125,133]. TSLP, IL-4, and IL-13 would accelerate the Th2-dominant environment [134]. As for the association of chemokines and their receptors, the infiltration of malignant T cells and the Th2-biased microenvironment are enhanced by eotaxins and HVEM from CAF, keratinocytes, and dendritic cells [129,133]. Furthermore, T_RM_ expressing CCR4, CCR8, and CCR10 can be recruited in the skin, attracted by CCL17, CCL18, CCL22, and CCL27, which are produced by the TAM, keratinocytes, and dendritic cells [123,124,127,135,136] (Figure 2).

Accordingly, TME in CTCL leads to the Th2-biased condition, affecting both malignant and benign T cells in the lesions, which would promote the expansion of malignant T cells and suppress the antitumor activities of benign T cells, sometimes making the distinction of these two fractions obscure.

## 8. Conclusions

Based on these findings, the characteristics of malignant and benign T cells in CTCL are summarized as below.

-The malignant T cells in MF typically possess the T_RM_ phenotype with a stronger reactive property to skin-derived IL-7 and/or IL-15.-Malignant T-cell population in MF consists of multiple subclones sometimes common between different lesions, suggesting that it might develop according to the repeated somatic mutations both before and after entering the skin.-While CD103^+^ CD8 T_RM_ reportedly contribute to the antitumor immunity with the production of IFNγ and granzymes in multiple solid cancers, benign T cells in CTCL lesions possess less T_RM_ phenotype with Th2-biased suppressive property.-As TME, the recruitment and proliferation of malignant T cells in skin can be supported by cytokines and chemokines in the skin. In addition, the related cells, such as TAM and CAF, are involved in the promotion of a Th2-biased microenvironment, affecting both malignant and benign T cells in CTCL.

Capturing the features of malignant and benign T cells in CTCL is necessary for understanding the pathogenesis of CTCL and would hopefully lead to new therapeutic strategies specifically targeting the skin malignant T cells or benign T cells.

## Figures and Tables

**Figure 1 ijms-22-12933-f001:**
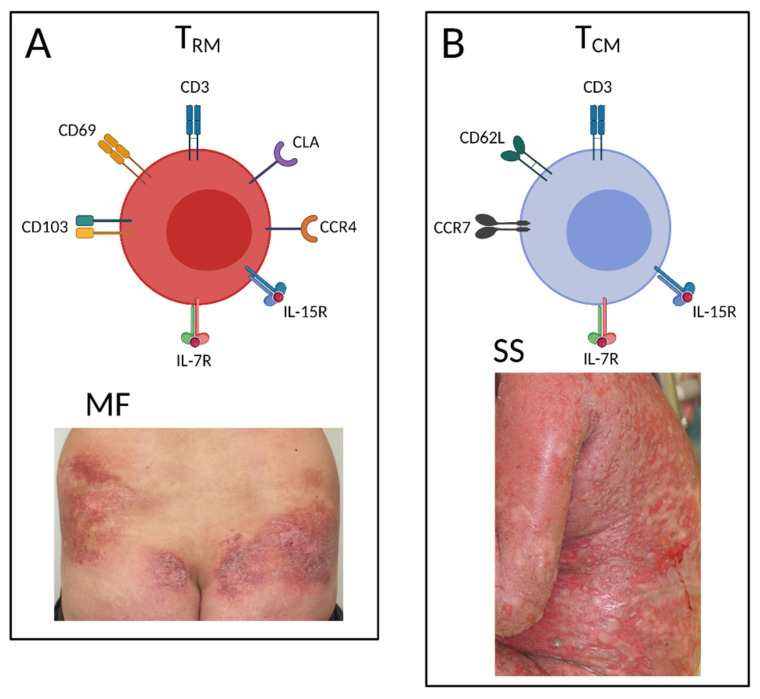
The cell-surface molecules characteristic of the malignant T cells found in mycosis fungoides (MF) and Sézary syndrome (SS). (**A**) The malignant T cells in a well-demarcated patch or plaque lesions of MF typically show the resident memory T cells (T_RM_) phenotype with CD69 and CD103 expression. (**B**) The malignant T cells found in diffuse erythema of SS typically show the central memory T cells (T_CM_) phenotype with CCR7 and CD62L expression. Created by BioRender.com.

**Figure 2 ijms-22-12933-f002:**
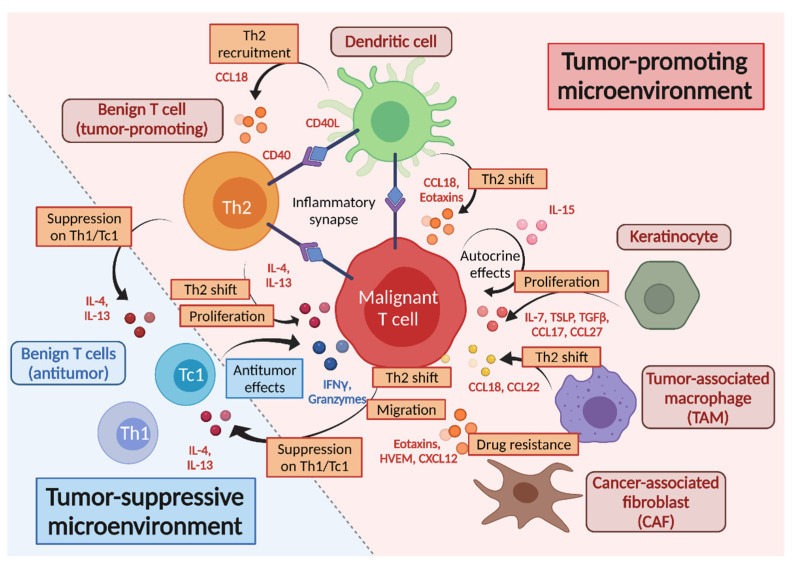
Tumor microenvironment in CTCL. The recruitment and proliferation of malignant T cells are promoted by chemokines and cytokines from the tumor-constituting cells, including TAM, CAF, dendritic cells, keratinocytes, and Th2 cells. IL-15 from the malignant T cells works in the autocrine manner too. Inflammatory synapses are formed among dendritic cells, benign Th2, and malignant T cells and contribute to Th2-biased microenvironment and tumor progression. Benign Th1 and Tc1 cells exert antitumor effects by producing IFNγ and granzymes, while they are suppressed by Th2 cytokines, such as IL-4 and IL-13, which benign Th2 and malignant T cells produce. Tumor-promoting molecules are represented in red, and tumor-suppressive molecules are indicated in blue.

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
