# Peer review of "Malignant and Benign T Cells Constituting Cutaneous T-Cell Lymphoma"

_ijms, 2021, doi:10.3390/ijms222312933_

Round 1

Reviewer 1 Report

The present review entitled “Malignant and benign T cells constituting cutaneous T-cell lymphoma” focuses on the features of both malignant and benign T cells, necessary for the better understanding of CTCL pathogenesis, leading to new therapeutic strategies specifically targeting the skin malignant T cells or benign T cells. The review highlights the malignant T cells resides in MF lesions are skin resident memory T cells (TRM) that resides in the peripheral tissues for long periods and represents the indolent nature of the disease. On the other hand, malignant T cells in SS are central memory T cells (TCM) which are characterized by recirculation to and from blood and lymphoid tissues. The review summarizes the property of skin TRM, and then provide the characteristics of malignant and benign T cells in MF/SS from the aspects of TRM and T-cell phenotypes.

An increased understanding of the malignant and benign T cells characteristics and phenotype may bring in a major breakthrough in its clinical application, resulting in more favorable and durable outcomes. I highly recommend this review. However, I do have some minor considerations which requires following addition to the manuscript.

  1. In tumor microenvironment in CTCL section, authors have nicely summarizes the effect of microenvironment on CTCL. However, there are couple of recent studies has been made which focuses on the role of CAFs in CTCL. Authors should discuss about the work done by Aronovich et al., 2021 (PMID: 32795528) which focuses on the role of CAFs in tumor cell migration and drug resistance in MF.
  2. Also, Mehdi et al., 2021 (PMID: 33941102) describes how normal and MF fibroblasts are phenotypically different and regulates TWIST1, TOX and cytokine gene expression in CTCL. Authors should also review this major finding which describes the indolent nature of disease.

Other than these minor modification, this review has been executed with care, and can be used by scientists since it provides adequate information specific for clinician as well as basic researchers, who wish to study malignant and benign T cells characteristics for CTCL treatment. Therefore, this review should be accepted for publication.

Author Response

Please see the attachment. Thank you so much for reviewing our manuscript.

Reviewer 2 Report

Dear authors,

I really enjoyed reading your paper. I think you have fully explored the subject so I have no major comments, only few rather cosmetic issues:

  1. In paragraph 5 please describe the immunophenotype of MF cells and Sezary cells.
  2. Figure 1: It would be more clear if you separate MF from SS with frames marked A and B
  3. Figure 2: A lot of information is presented in the figure, please insert annotations to figure 2 in the text where the mechanism presented in the figure is described, I would be more clear to the reader.

Author Response

(The authors gave the same response as above.)
